# Systematic Failures in Collective Reasoning under Distributed Information in Multi-Agent LLMs

Yuxuan Li [1]  Aoi Naito [1 2 3]  Hirokazu Shirado [1]

## Abstract

Multi-agent systems built on large language models (LLMs) are expected to enhance decision-making by pooling distributed information, yet systematically evaluating this capability has remained challenging. We introduce HIDDEN-BENCH, a 65-task benchmark grounded in the Hidden Profile paradigm, which isolates collective reasoning under distributed information from individual reasoning ability. Evaluating 15 frontier LLMs, we find that multi-agent LLMs achieve only 30.1% accuracy under distributed information, compared to 80.7% accuracy for single agents given complete information. We trace this gap to a systematic failure mode: agents cannot recognize or act under latent information asymmetry—they fail to reason about what others might know but have not yet expressed, leading to premature convergence on shared evidence while critical distributed facts remain unexplored. These failures persist across prompting strategies, communication depths, and group sizes—and worsen as groups scale. While some models (e.g., Gemini-2.5-Flash/Pro) outperform others, neither model scale nor individual reasoning accuracy reliably predicts collective performance. We further show that this bottleneck is actionable: a lightweight structured communication protocol substantially improves collective reasoning across model families. Our results identify failures in collective information exploration in decision-making as a key limitation of multi-agent LLMs, and provide a theory-grounded, reproducible framework for diagnosing collective reasoning failures.

[1]School of Computer Science, Carnegie Mellon University, Pittsburgh, USA [2]School of Environment and Society, Institute of Science Tokyo, Tokyo, Japan [3]Japan Society for the Promotion of Science, Tokyo, Japan. Correspondence to: Yuxuan Li <yuxuanll@andrew.cmu.edu>.

*Proceedings of the $43^{rd}$ International Conference on Machine Learning*, Seoul, South Korea. PMLR 306, 2026. Copyright 2026 by the author(s).

## 1. Introduction

Consider a team of AI agents diagnosing a complex system failure. One agent monitors network logs, another hardware sensors, and a third software exceptions. Each agent holds partial evidence about the root cause. The correct diagnosis requires integrating all three perspectives. Yet even in this seemingly straightforward setting, will the agents successfully pool their distributed information to reach a better decision than any single agent?

This scenario illustrates a central challenge in multi-agent systems built on large language models (LLMs): *collective reasoning under distributed information* (Stasser & Titus, 1985; Schulz-Hardt & Mojzisch, 2012; Woolley et al., 2010). Multi-agent LLM systems are increasingly deployed for tasks requiring collaboration, diverse perspectives, and distributed expertise (Li et al., 2023; Qian et al., 2024; Hong et al., 2023; Dong et al., 2024; Piao et al., 2025). Their promise rests on the assumption that groups of agents can integrate more information than any single agent alone (Du et al., 2024; Zhang et al., 2024; Pan et al., 2024; Liu et al., 2023). However, effective collective decision-making requires coordinating *information exploration*—eliciting potentially relevant information held by other agents—with *information integration*—combining available evidence to reach a decision (March, 1991; Hills et al., 2015; Dimakopoulou & Van Roy, 2018).

While recent studies suggest that multi-agent LLM systems face coordination challenges (Jones & Steinhardt, 2022; Shi et al., 2024; Sumita et al., 2024; Cemri et al., 2025a), it remains unclear how such failures arise from limitations in multi-agent interaction. A key difficulty is evaluation. Without controlled information distribution and access to ground truth, it is difficult to evaluate whether multi-agent deliberation improves decision accuracy beyond what individual agents can achieve alone.

The *Hidden Profile paradigm*, originally developed in social psychology (Stasser & Titus, 1985; Schulz-Hardt & Mojzisch, 2012), provides a controlled way to isolate this coordination problem. In Hidden Profile tasks, each group member holds unique information that must be pooled to reach the correct decision, while shared information favors

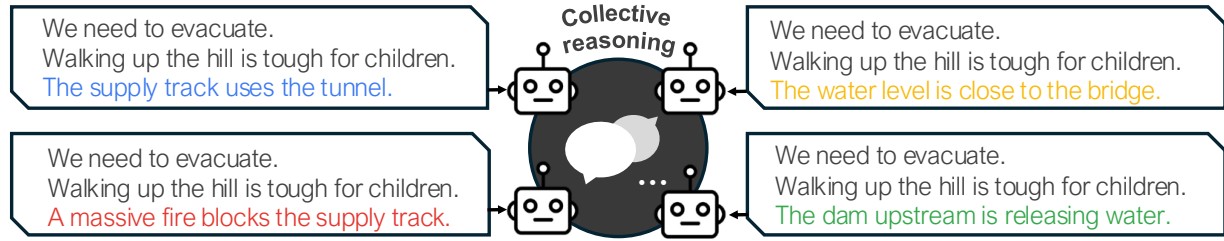

*Figure 1.* Overview of the Hidden Profile paradigm. Agents receive shared information (black) and unshared information (color) without recognizing the asymmetry. Only by sharing unshared information can they identify the optimal decision—here, walking up the hill rather than taking the other options (the tunnel and the bridge). See Table 1 for the actual information distribution.

an incorrect alternative (Fig. 1). Critically, individuals do not know in advance which information is not shared or relevant to the correct decision. Because each task becomes solvable once the distributed information is surfaced, the paradigm enables direct evaluation of whether multi-agent deliberation affects decision accuracy.

Grounded in this paradigm, we develop HIDDENBENCH, a scalable and reproducible benchmark that disentangles collective reasoning failures from individual reasoning ability by construction. Across 65 tasks, success requires pooling distributed information, while each task remains solvable by a single agent given complete information.

Using HIDDENBENCH, we evaluate 15 frontier LLMs under controlled multi-agent communication settings. We find that multi-agent LLMs achieve only 30.1% accuracy when information is distributed, compared to 80.7% accuracy for single agents given complete information. This gap persists across prompting strategies, communication depths, and group sizes, and worsens as groups scale. While some models perform better than others, neither model scale nor individual reasoning accuracy reliably predicts collective reasoning performance.

Through targeted ablations, we identify systematic failure modes in multi-agent reasoning under distributed information. Agents can integrate information once disclosed, but fail to recognize and act on latent information asymmetry—converging on shared evidence without actively eliciting unobserved knowledge held by others. Our findings suggest that improving multi-agent performance requires coordination mechanisms and learning objectives that incentivize epistemic exploration, rather than relying solely on scaling individual reasoning.

Our contributions are fourfold.

- We introduce HIDDENBENCH, a scalable and reproducible 65-task benchmark that isolates collective reasoning under distributed information while controlling for individual reasoning ability.

- Across 15 frontier models, we identify a systematic coordination failure in which agents fail to surface critical unshared information despite being able to integrate it once disclosed.
- We show that failures in collective information exploration persist across prompting strategies, communication depths, and model families, while worsening as the number of agents increases.
- We demonstrate that the identified bottleneck is actionable through a lightweight communication protocol that substantially improves collective reasoning.

## 2. Related Work

### 2.1. Assessing Multi-Agent LLM Systems

Recent advances have spurred interest in multi-agent LLMs, where models interact through dialogue or coordination to solve complex tasks collectively (Li et al., 2023; Du et al., 2024; Qian et al., 2024; Guo et al., 2024; Zhang et al., 2023; Wang et al., 2024; Huang et al., 2025). Applications range from software development (Wu et al., 2024; Qian et al., 2023; Hong et al., 2023; Dong et al., 2024; Antoniades et al., 2024) to scientific discovery (Zheng et al., 2023; Schmidgall et al., 2025; Boiko et al., 2023; Swanson et al., 2024) and social simulation (Park et al., 2023; Piao et al., 2025; Gao et al., 2023; Xie et al., 2024a).

The central assumption is that groups of LLMs can be more robust and diverse than single models (Du et al., 2024; Qian et al., 2024; Zhang et al., 2024; Pan et al., 2024; Liu et al., 2023; Wang et al., 2024). However, there lacks theory-driven frameworks that cleanly separate failures of individual reasoning from failures of collective information integration (Li et al., 2023; Schmidgall et al., 2025; Gong et al., 2023; Abdelnabi et al., 2023; Zhou et al., 2023; Cemri et al., 2025b; Raileanu et al., 2018; Wang et al., 2020). Existing evaluations typically conflate collective reasoning with task-specific skills, coordination protocols, or domain knowledge (Sun et al., 2025; Xu et al., 2024; Zhu et al., 2025; Bu et al., 2025; Xie et al., 2024b)—none isolate infor-

mation asymmetry as a controlled variable. Closely related are studies of transfer mechanisms underlying multitasking and shared representations in agent learning (Wu et al., 2020; Zhang et al., 2026; Yang et al., 2025), which examine when and how information generalizes across tasks. Our work extends this line by introducing a formalized, theory-grounded benchmark that isolates collective reasoning under distributed information from individual reasoning ability rather than optimizing task performance.

## 2.2. Collective Reasoning Failures in Human Groups

Social psychology shows that communication can suppress rather than improve group performance (Kerr & Tindale, 2004; Janis, 1972; Lorenz et al., 2011; Muchnik et al., 2013). Failures often arise when groups neglect unique knowledge (shared information bias) (Stasser & Titus, 1985; Schulz-Hardt & Mojzisch, 2012; Toma & Butera, 2009), conform to majorities (conformity bias) (Asch, 1956; Moscovici & Faucheux, 1972; Leibenstein, 1950), adhere to prevailing social norms (social desirability bias) (Fisher, 1993; Mahmoodi et al., 2015), or favor the status quo (normalcy bias) (Drabek, 2012; Shirado et al., 2020), regardless of their veracity. These dynamics can culminate in over-coordination, entrenched beliefs, or groupthink (Nwana et al., 2005; Gulati et al., 2012; Shirado & Christakis, 2017; Chang et al., 2017; Park et al., 2010; Janis, 1972; McCauley, 1989; Park, 2000).

While these failures are well-documented in humans, their emergence in multi-agent LLMs remains poorly characterized. A key open question is whether the informational structure that challenges human groups also constrains multi-agent AI systems. Our study addresses this by adapting the Hidden Profile paradigm (Stasser & Titus, 1985; Schulz-Hardt & Mojzisch, 2012; Toma & Butera, 2009)—a canonical diagnostic for human group failures—into a reproducible benchmark for LLM agents. The paradigm operationalizes longstanding theoretical concerns from distributed knowledge representation (Fagin et al., 2004), transactive memory (Wegner, 1987), and the exploration–exploitation trade-off in organizations (March, 1991), where coordinated information surfacing—not just inference—is central to group performance.

## 3. Hidden Profile Tasks for Multi-Agent LLMs

### 3.1. Task setup

We study collective reasoning under distributed information using tasks adapted from the *Hidden Profile paradigm* in social psychology. In these tasks, no agent can identify the correct option from its local information alone, while the correct decision becomes attainable only once distributed information is pooled by multi-agent interaction through multiple rounds of communication. The same coordination challenge arises across many real-world settings—medical diagnosis with multiple specialists (Centola et al., 2023), incident response across distributed logs (Cichonski et al., 2012), and intelligence analysis with complementary partial evidence (Crowther, 2014)—where critical evidence is naturally scattered across team members. We adopt this paradigm as a principled choice for our setting: it directly targets reasoning over distributed information, provides a verifiable ground truth, and is well-established in decades of human studies—giving us confidence that diagnoses transfer to real-world settings, where ground truth is often ambiguous and coordination failures are difficult to attribute.

Each task consists of a set of decision options and a collection of task-relevant facts. Under the *Hidden Profile* condition, a subset of facts is shared among all agents, while the remaining facts are uniquely distributed across agents. The shared information is constructed to favor an incorrect option, whereas the unshared information—when combined—supports the correct one (Fig. 1). In contrast, under the *Full Profile* condition, all agents receive the full set of task-relevant information, including the critical facts, from the outset. Agents are not informed whether their information differs from that of others.

This design isolates failures in *multi-agent interaction* from limitations in *individual reasoning*. Each task is solvable by a single agent under the Full Profile condition, providing an upper bound on achievable performance. Conversely, low pre-discussion accuracy under the Hidden Profile condition ensures that success requires information sharing rather than chance. Accordingly, we evaluate post-discussion accuracy of multi-agent systems under the Hidden Profile condition and compare it against single-agent accuracy under the Full Profile condition. A formal specification of the task structure, decision rules, and success criteria is provided in Appendix A.2.

### 3.2. Verification

We instantiate three Hidden Profile tasks (Table 1) and verify that the construction satisfies the defining properties of collective reasoning under distributed information using both human groups and GPT-4.1 agents. Details of the human studies are provided in Appendix A.3.

Human groups satisfy the Hidden Profile condition: pre-discussion accuracy is low under distributed information ($Y^{\text{pre}} = 0.125$) but substantially higher under the Full Profile condition ($Y^{\text{full}} = 0.604$). Discussion improves performance under the Hidden Profile condition, yet post-discussion accuracy remains below the Full Profile ceiling ($Y^{\text{post}} = 0.385 < Y^{\text{full}}$, $p = 0.003$), consistent with prior findings.

*Table 1.* Example Hidden Profile task used for verification, where agents choose among North Hill, East Town, and West City, and the correct decision is North Hill. Seven shared facts $\mathcal{I}_s = \{s_1, \ldots, s_7\}$ are available to all agents ($a_1$–$a_4$), while four critical facts $\mathcal{I}_u = \{u_1, \ldots, u_4\}$ are uniquely distributed such that agent $a_i$ receives $I_i = \mathcal{I}_s \cup \{u_i\}$ under the Hidden Profile condition. Under the Full Profile condition, all agents receive the complete information set $\mathcal{I}_s \cup \mathcal{I}_u$. The table shows abbreviated summaries of task facts for readability; agents receive the full natural-language statements during experiments.

| ID | Type | Statement Summary | $a_1$ | $a_2$ | $a_3$ | $a_4$ |
|---|---|---|---|---|---|---|
| $s_1$ | Shared | West City is accessible via a bridge over the river. | ✓ | ✓ | ✓ | ✓ |
| $s_2$ | Shared | East Town is accessible via a tunnel on middle ground. | ✓ | ✓ | ✓ | ✓ |
| $s_3$ | Shared | North Hill is accessible via driveway and walking trails. | ✓ | ✓ | ✓ | ✓ |
| $s_4$ | Shared | West City hotels are ready with supplies. | ✓ | ✓ | ✓ | ✓ |
| $s_5$ | Shared | East Town offers shelter and volunteers. | ✓ | ✓ | ✓ | ✓ |
| $s_6$ | Shared | North Hill school is usable but lacks privacy. | ✓ | ✓ | ✓ | ✓ |
| $s_7$ | Shared | Mudslide blocks walking trails to North Hill. | ✓ | ✓ | ✓ | ✓ |
| $u_1$ | Unshared | River level is just below the bridge. | ✓ | | | |
| $u_2$ | Unshared | Dam upstream will release water in a minute. | | ✓ | | |
| $u_3$ | Unshared | Supply truck was heading to the tunnel. | | | ✓ | |
| $u_4$ | Unshared | Massive fire blocks the supply truck. | | | | ✓ |

GPT-4.1 agents exhibit a similar pattern. Pre-discussion accuracy under the Hidden Profile condition is near zero ($Y^{\text{pre}} = 0.008$), while accuracy under the Full Profile condition is high ($Y^{\text{full}} = 0.733$), confirming that the tasks are individually solvable given full information. Communication improves performance ($0.008 \rightarrow 0.233$, $p < 0.001$; Fisher's exact test), but post-discussion accuracy remains far below the Full Profile upper bound.

In particular, GPT-4.1 agents converge primarily on shared information while failing to actively reason about and surface unshared evidence. These results verify that tasks instantiated from our construction induce genuine collective reasoning challenges. They also show that observed multi-agent LLM failures cannot be attributed to task difficulty or insufficient individual reasoning capability, as agents achieve high accuracy under the Full Profile condition. Instead, failures under the Hidden Profile condition reflect limitations in decentralized information exploration under latent information asymmetry.

## 4. HIDDENBENCH: A Scalable Benchmark for Collective Reasoning

Given the consistent failures observed in our initial Hidden Profile tasks, we construct HIDDENBENCH as a diagnostic benchmark for systematically characterizing limitations in multi-agent LLM systems. Rather than targeting task difficulty or individual reasoning ability, HIDDENBENCH is designed to isolate failures arising from decentralized coordination under distributed and partially observable information. We release the full benchmark, customizable evaluation suite, and generated corpus via Hugging Face (Face, 2026) and GitHub (GitHub, 2026).

To construct a scalable benchmark, we extend beyond established tasks from social psychology to automatically generated ones with theory-based verification. As a result, HIDDENBENCH consists of 65 Hidden Profile tasks spanning diverse decision-making contexts with varying information structures (e.g., healthcare, organizational planning, cultural preservation).

### 4.1. Adaptations from Human Studies

We systematically reviewed studies summarized in a major Hidden Profile meta-analysis (Lu et al., 2012) and identified all publicly available task materials. From this review, we selected and adapted five scenarios from prior literature (Stasser & Stewart, 1992; Graetz et al., 1998; Toma & Butera, 2009; Baker, 2010; Schulz-Hardt & Mojzisch, 2012) that demonstrated robust Hidden Profile effects in human experiments. Each adapted task preserves the original information structure and decision options while standardizing the format for multi-agent LLM evaluation. We maintained the original distribution of shared versus unshared information and ensured that the correct decision could only be identified through successful integration of distributed knowledge. All adapted items were validated against the formal model defined in Appendix A.2.

### 4.2. Automatic Pipeline for Scalable Task Generation

To scale beyond manually crafted and adapted tasks, we developed an automatic generation pipeline that produces Hidden Profile scenarios with the defined system structure. The pipeline operates in three stages: generation, execution, and selection (Fig. 2).

In the generation stage, GPT-4.1 is prompted to create novel Hidden Profile tasks following a structured template. Each

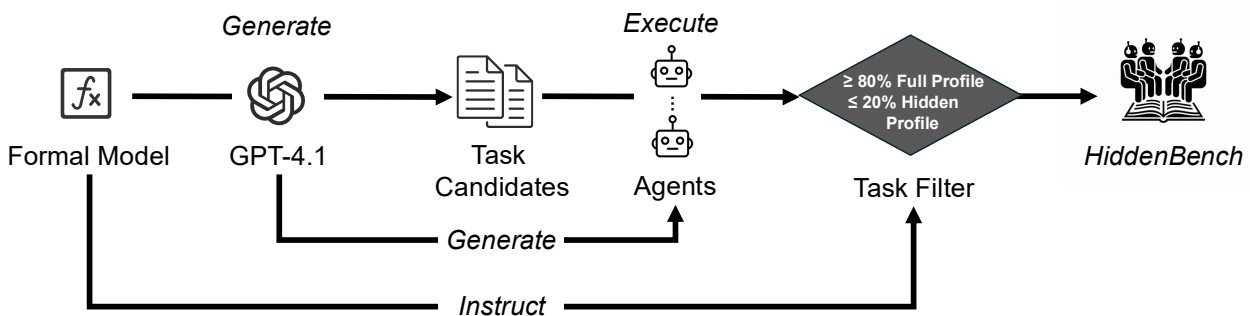

*Figure 2.* Automatic pipeline for scalable Hidden Profile task generation. GPT-4.1 generates candidate tasks, which are then tested under both Full and Hidden Profile conditions across 10 sessions each. Tasks that satisfy validation thresholds ($\geq 80\%$ pre-discussion accuracy in the Full Profile condition; $\leq 20\%$ in the Hidden Profile condition) are retained in HiddenBench. From 200 candidates, the pipeline produced 57 validated tasks (28.5% validation rate).

task includes (1) a scenario description with clear decision options, (2) shared information available to all agents, (3) unshared information distributed among agents, and (4) a designated correct answer that requires integrating both shared and unshared information.

In the execution stage, each generated task is executed in two conditions. In the Full Profile condition, agents receive all information (shared + unshared), allowing individual identification of the correct answer. In the Hidden Profile condition, each agent receives only shared information plus their unique unshared pieces, enforcing the Hidden Profile constraint. We run 10 simulation sessions per condition with GPT-4.1 agents and measure pre-discussion decision accuracy without any inter-agent interaction.

In the selection stage, tasks pass only if they meet two criteria: high accuracy ($\geq 80\%$) in the Full Profile condition, ensuring task solvability, and low accuracy ($\leq 20\%$) in the Hidden Profile condition, confirming that no agent can succeed without information aggregation. Both thresholds refer to per-agent pre-discussion accuracy averaged across 10 sessions, measured before any inter-agent interaction. This filtering ensures that each task creates a genuine Hidden Profile scenario requiring collective reasoning.

From 200 candidates, 57 tasks passed validation (28.5% validation rate). Combined with three manually designed tasks and five adapted from prior studies, HIDDENBENCH comprises 65 scenarios in total. The pipeline is fully reproducible and can be extended to generate additional validated tasks as needed.

### 4.3. Benchmark Extensibility

Collective reasoning under distributed information depends strongly on domain semantics: whether agents recognize which facts matter, how information relates to decision options, and what counts as sufficient evidence all vary across

contexts. Accordingly, HIDDENBENCH is designed not only as a fixed benchmark suite, but also as a reproducible pipeline for generating tasks with controlled distributed-information structures.

By enforcing verification criteria ($\geq 80\%$ Full Profile accuracy, $\leq 20\%$ Hidden Profile accuracy), the pipeline ensures each generated task isolates coordination capability from individual reasoning ability. This design enables principled extensions across domains, group sizes, and communication protocols, positioning HIDDENBENCH as a reusable framework for studying coordination failures in multi-agent systems.

## 5. Assessing Collective Reasoning with HIDDENBENCH

We use HIDDENBENCH to characterize systematic failure modes in collective reasoning across state-of-the-art multi-agent LLM systems. We evaluate 15 frontier models spanning four families—OpenAI GPT, Google Gemini, Alibaba Qwen, and Meta Llama—under controlled Hidden and Full Profile conditions. For each model, we run 10 sessions per task and measure accuracy before and after multi-agent interaction using the average rule (i.e., accuracy before and after discussion).

Figure 3 reveals a consistent failure pattern across all models. Under the Hidden Profile condition, pre-discussion accuracy remains uniformly low (0.082–0.217), indicating that agents cannot identify the correct option from local information alone. In contrast, under the Full Profile condition, pre-discussion accuracy ranges from 0.435 to 0.981, with most state-of-the-art models exceeding 0.8. This confirms that the same tasks are individually solvable when complete information is available and that failures under the Hidden Profile condition cannot be attributed to insufficient individual reasoning capability.

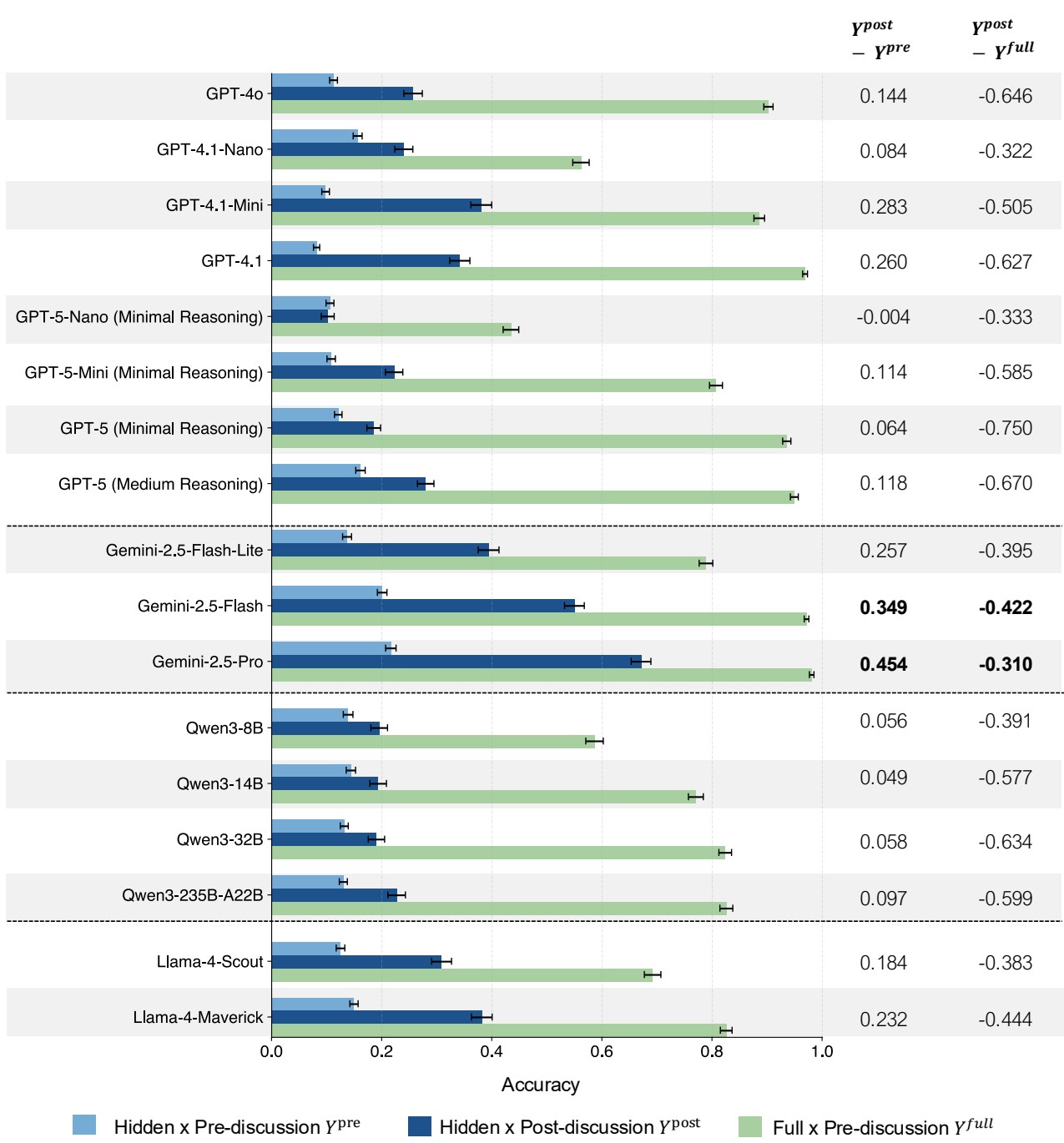

*Figure 3.* Collective reasoning performance across 15 LLMs on HIDDENBENCH. Bars show average accuracy across 65 tasks under the average rule. The rightmost columns display the improvement from interaction ($Y^{\text{post}} - Y^{\text{pre}}$) and the gap between collective reasoning and individual reasoning with full information ($Y^{\text{post}} - Y^{\text{full}}$). Models meeting strong collective reasoning criteria ($Y^{\text{full}} > 0.8$ and $Y^{\text{post}} - Y^{\text{pre}} > 0.4 \times (Y^{\text{full}} - Y^{\text{pre}})$) are highlighted in bold. Error bars indicate mean $\pm$ s.e.m.

Despite strong individual performance under the Full Profile condition, collective performance remains systematically limited. Post-discussion accuracy under the Hidden Profile condition improves relative to pre-discussion baselines, showing that inter-agent interaction enables partial integration of distributed information. However, the magnitude of improvement varies widely across models—from negligible (GPT-5-Nano: –0.004) to substantial (Gemini-2.5-Pro: 0.454). Crucially, even the strongest models fall well short of their own Full Profile pre-discussion performance, with persistent gaps ranging from –0.310 (Gemini-2.5-Pro) to –0.750 (GPT-5 Minimal Reasoning). This pattern indicates a structural limitation in multi-agent reasoning under latent information asymmetry, even when individual reasoning is strong.

The benchmark further reveals systematic differences in how models fail. For example, the Gemini family consistently outperforms other model families in collective settings. Gemini-2.5-Pro achieves the highest post-discussion accuracy under the Hidden Profile condition (0.671) and the smallest gap relative to Full Profile performance (–0.310), with Gemini-2.5-Flash (0.550) and Gemini-2.5-Flash-Lite (0.394) also performing competitively. In contrast, improvements in model scale or individual reasoning ability do not consistently translate into stronger collective reasoning. Despite enhanced reasoning performance under the Full Profile condition, GPT-5 variants fail to substantially outperform smaller models such as GPT-4.1-Mini in multi-agent settings.

Taken together, these results reveal a persistent gap between individual reasoning ability and collective reasoning under distributed information. Although communication improves performance, current multi-agent LLMs do not reliably surface and coordinate unshared information under latent asymmetry. HIDDENBENCH exposes this limitation while providing a controlled framework for evaluating mechanisms that improve collective information exploration.

# 6. Ablation Studies: Diagnosing Failures in Multi-Agent Coordination

We conduct targeted ablations to assess the robustness of collective reasoning failures and to isolate the primary bottleneck underlying these failures. All experiments use GPT-4.1 as a representative strong baseline on three tasks from HIDDENBENCH, unless otherwise noted. Through these ablations, we show that the dominant failure is not aggregation or inference, but action selection: agents do not reliably choose communicative actions that maximize expected information gain about unobserved task-relevant variables.

*Table 2.* Effect of communication depth (GPT-4.1). Performance peaks at $T = 15$ but remains far below Full Profile.

| Rounds | $Y^{\text{post}}$ | Improvement | Gap to $Y^{\text{full}}$ |
|--------|---------|-------------|------------|
| 5 | $0.108 \pm 0.058$ | +0.100 | −0.625 |
| 10 | $0.200 \pm 0.100$ | +0.183 | −0.533 |
| 15 | $0.233 \pm 0.069$ | +0.225 | −0.500 |
| 20 | $0.133 \pm 0.033$ | +0.125 | −0.600 |

*Table 3.* Effect of prompting strategies (GPT-4.1). No strategy resolves collective reasoning failures.

| Strategy | Average | Majority |
|----------|---------|----------|
| Very Cooperative | 0.242 | 0.233 |
| Cooperative | 0.200 | 0.200 |
| Constructive | 0.200 | 0.167 |
| Conflictual | 0.017 | 0.000 |
| Very Conflictual | 0.258 | 0.000 |
| Zero-shot CoT | 0.222 | 0.222 |
| Informing Asymmetry | 0.367 | 0.367 |
| Share All Information | 0.467 | 0.467 |

## 6.1. Failures Persist Across Experimental Variations

**Communication Depth.** We vary the number of communication rounds $T \in \{5, 10, 15, 20\}$ (Table 2). Performance peaks at $T = 15$ but declines at $T = 20$, suggesting extended discussion reinforces incorrect consensus rather than promoting exploration. Even at optimal depth, post-discussion accuracy (0.233) remains far below Full Profile (0.733).

**Prompting Strategies.** We next test whether prompting can steer agents toward better coordination (Table 3), including cooperation–conflict styles, zero-shot chain-of-thought (Wei et al., 2022), explicit asymmetry awareness, and direct instructions to share information. No prompting strategy substantially closes the gap to Full Profile performance. Cooperative prompts yield $Y^{\text{post}} = 0.200$–$0.242$. Conflictual prompts fail to converge, producing zero majority consensus. Explicit instructions to share all information improve accuracy to 0.467, but still leave roughly half the gap unresolved, indicating that disclosure alone is insufficient without mechanisms for identifying and prioritizing critical unshared evidence [1].

[1] Note that Share All Information is a prompting intervention requiring agents to disclose voluntarily through dialogue; this differs from Reveal-All in Section 6.3, a mechanistic intervention that appends all information available to an agent to its round-1 message; and from Full Profile 3.1, which provides complete information to an agent from the outset.

*Table 4.* Effect of group size (GPT-4.1). Failures worsen as $N$ increases.

| $N$ | # Tasks | $Y^{\text{pre}}$ | $Y^{\text{post}}$ | Improvement |
|-----|---------|------------------|-------------------|-------------|
| 3 | 7 | 0.167 | 0.514 | +0.348 |
| 4 | 58 | 0.072 | 0.321 | +0.250 |
| 5 | 5 | 0.116 | 0.180 | +0.064 |
| 6 | 5 | 0.087 | 0.283 | +0.197 |
| 7 | 5 | 0.017 | 0.023 | +0.006 |

**Group Size.** We vary the number of agents $N \in \{3, 4, 5, 6, 7\}$, generating new validated tasks for $N > 4$. As shown in Table 4, failures *worsen* as groups scale: improvement from communication drops from +0.348 at $N = 3$ to +0.006 at $N = 7$, despite near-perfect Full Profile accuracy. This pattern poses a practical concern: multi-agent systems are often scaled precisely to pool diverse information, yet without explicit coordination mechanisms, adding agents amplifies failure rather than improving collective performance (Dimakopoulou & Van Roy, 2018).

## 6.2. Structured Dissent as a Minimal Coordination Mechanism

Real-world collective intelligence often relies on mechanisms that prevent premature consensus and encourage epistemic challenge (Jaques et al., 2019; Li et al., 2025). We evaluate heterogeneous four-agent groups across five frontier models, varying the proportion of agents prompted to adopt an adversarial, dissenting stance from 0% to 100%. The effect of dissent on collective reasoning is highly heterogeneous (Table 5).

For GPT-4.1, introducing a single adversarial agent (25% dissent) nearly doubles post-discussion accuracy (0.233 → 0.492), revealing a Goldilocks effect: minimal epistemic pressure disrupts premature convergence, but adding more adversaries impairs convergence on any decision. GPT-5-Mini shows a similar inverted-U with a broader peak at 50–75% dissent, and notably collapses to zero accuracy without any adversarial pressure. GPT-4.1-Mini benefits roughly monotonically from additional dissent (0.333 → 0.583), while Gemini-2.5-Flash-Lite is relatively stable across compositions, peaking at 75% (0.778) before dropping at 100%.

Simple dissent *hurts* Gemini-2.5-Flash: the model achieves its highest accuracy (0.889) without any adversarial agents and degrades sharply with even a single one (0.306). This suggests that when a model already coordinates effectively without intervention, imposed adversarial pressure can disrupt rather than support information surfacing.

Across all five models, even the best composition typically falls short of Full Profile performance, and there is no uni-

*Table 5.* Effect of group composition across five frontier models. Values are post-discussion accuracy ($Y^{\text{post}}$) for four-agent groups with 0–100% adversarial agents; bold marks the best composition for each model.

| Model | 0% | 25% | 50% | 75% | 100% |
|-------|------|------|------|------|------|
| *GPT* | | | | | |
| 4.1 | 0.233 | **0.492** | 0.342 | 0.242 | 0.375 |
| 4.1-mini | 0.333 | 0.361 | 0.556 | 0.528 | **0.583** |
| 5-mini | 0.000 | 0.278 | **0.361** | **0.361** | 0.250 |
| *Gemini-2.5* | | | | | |
| Flash | **0.889** | 0.306 | 0.361 | 0.222 | 0.278 |
| Flash-Lite | 0.667 | 0.667 | 0.639 | **0.778** | 0.500 |

*Table 6.* Isolating the bottleneck on 18 HIDDENBENCH tasks. Forced disclosure nearly eliminates failure across both models; passive summarization yields more limited and model-dependent gains.

| Condition | GPT-4.1 | Gemini-2.5-Flash |
|-----------|---------|------------------|
| Baseline | 0.037 | 0.173 |
| Reveal-All | 0.926 | 0.982 |
| Secretary | 0.241 | 0.713 |

versally beneficial level of dissent. Thus, while structured dissent can mitigate coordination failures for some models, heterogeneity alone is insufficient to resolve reasoning under latent information asymmetry, and the most effective intervention, if any, depends on the underlying model. Robust collective reasoning will require more explicit coordination mechanisms that support systematic information surfacing and integration.

## 6.3. Isolating the Bottleneck

Where in collective reasoning do agents fail: sharing information or integrating it? We design two interventions to isolate each component:

- **Reveal-All**: Each agent discloses all information in the first round before discussion, removing the sharing bottleneck.
- **Secretary**: A dedicated agent summarizes all disclosed information each round, testing whether passive repetition aids integration.

We evaluate both interventions on 18 HIDDENBENCH tasks (3 manually crafted and 15 randomly sampled) using GPT-4.1 and Gemini-2.5-Flash as representatives of two model families.

Table 6 reveals a clear pattern. When forced to disclose all facts, accuracy rises to 92.6% (GPT-4.1) and 98.2% (Gemini-2.5-Flash), nearly matching Full Profile—demonstrating that agents *can* integrate information once

they have it. This rules out limitations in individual reasoning, memory, or aggregation as primary failure sources; the failure lies in surfacing information, not reasoning over it. Passive summarization yields more limited and model-dependent gains: GPT-4.1 shows little benefit (0.241), while Gemini-2.5-Flash improves substantially (0.713), suggesting that integration ability also varies across models. Importantly, although forced disclosure nearly eliminates failure, the space of task-relevant information is itself latent and must be discovered through interaction. Effective collective reasoning therefore requires *active probing* and recognition of information asymmetry, not mere repetition.

Qualitative analysis supports this interpretation. Top-performing models (e.g., Gemini-2.5-Pro) signal asymmetry ("Your point about the dam is new to me"), actively probe for missing facts, and offer reasoned disagreement. In contrast, lower-performing models converge prematurely ("I agree—let's choose East Town") before hidden facts surface. These behaviors indicate that collective failure arises from missing mechanisms for detecting and managing latent information asymmetry during interaction, rather than from deficiencies in individual reasoning or aggregation.

### 6.4. Structured Communication as a Coordination Mechanism

The preceding ablations do not yet test whether HIDDEN-BENCH can guide the design of improved coordination mechanisms. To test this, we evaluate a lightweight structured communication protocol that targets the two failure modes identified above—failure to surface unshared information and premature convergence—without explicitly informing agents of asymmetry or requiring full disclosure.

The protocol consists of two stages. In the *Exchange* stage (2 rounds), each agent shares 1–2 decision-relevant facts and gives one reason the current front-runner may be incorrect. In the *Decide* stage (1 pass), each agent summarizes the strongest evidence and remaining uncertainty before voting. Importantly, the protocol does not assume prior knowledge of which information is missing or relevant, preserving the core coordination challenge of the Hidden Profile setting.

We evaluate this protocol on 18 HIDDENBENCH tasks (3 manually crafted + 15 randomly sampled, same as in Section 6.3; 5 runs each, 4 agents) across three models. As shown in Table 7, the protocol substantially improves decision accuracy across all three. While the protocol is intentionally lightweight and does not resolve all coordination challenges, the consistent improvement across model families demonstrates that HIDDENBENCH is *actionable*: it can identify and evaluate coordination structures that meaningfully improve collective reasoning under distributed information.

*Table 7.* Structured communication protocol on 18 HIDDEN-BENCH tasks. A lightweight Exchange-then-Decide protocol substantially improves collective reasoning across three model families, without revealing the information asymmetry or requiring full disclosure.

| Model | Baseline | Structured |
|---|---|---|
| GPT-4.1 | 0.037 | 0.800 |
| Gemini-2.5-Flash | 0.173 | 0.727 |
| Gemini-2.5-Flash-Lite | 0.043 | 0.743 |

## 7. Conclusion

We studied collective reasoning under distributed information in multi-agent LLM systems through the lens of the Hidden Profile paradigm. We introduced HIDDENBENCH, a 65-task benchmark that isolates failures in collective information exploration under distributed information from individual reasoning ability. Across 15 frontier LLMs, we observed a large and persistent gap between multi-agent performance with distributed information and single-agent performance with full information.

This gap persists across prompting strategies, communication depths, and group sizes. While agents can integrate information once it is disclosed, they consistently fail to surface critical unshared facts during interaction, indicating a coordination failure driven by latent information asymmetry. Building on this diagnosis, we showed that a lightweight structured communication protocol substantially improves performance across multiple model families, demonstrating that HIDDENBENCH can also serve as a testbed for developing coordination mechanisms.

Our findings suggest that improving collective reasoning under distributed information requires mechanisms that incentivize epistemic exploration—actively probing for missing evidence and prioritizing unshared information—rather than relying solely on stronger individual reasoning or longer interaction. HIDDENBENCH provides a scalable and reproducible framework for diagnosing these failures and measuring progress toward multi-agent systems capable of reliable collective reasoning under distributed information.

## Acknowledgements

This work was supported by the NOMIS Foundation and JSPS KAKENHI (Grant Number JP23KJ0879).

## Impact Statement

This paper presents work whose goal is to advance the understanding and evaluation of collective reasoning under distributed information in multi-agent systems built on large language models. By introducing a theory-grounded benchmark and identifying systematic failure modes under distributed information, this work aims to support the development of more reliable, transparent, and effective collaborative AI systems.

Potential societal impacts are primarily indirect and methodological. Improved evaluation of collective reasoning may inform the safer deployment of multi-agent AI in settings such as decision support, scientific collaboration, and organizational planning. At the same time, our findings highlight current limitations that caution against over-reliance on such systems in high-stakes or safety-critical contexts without additional safeguards.

The human subjects experiments reported in this paper were conducted with approval from an Institutional Review Board (IRB). We do not anticipate direct negative societal impacts arising from this work. The benchmark and analyses are intended for research and diagnostic purposes, and do not introduce new deployment mechanisms or applications. Overall, we believe the ethical and societal implications of this work are aligned with those commonly associated with foundational research in machine learning.

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

# A. Appendix

## A.1. LLM Usage

Except for the study itself, which directly evaluates LLM capabilities, we used LLMs solely to polish the writing of this manuscript and not for any other purpose.

## A.2. Formalizing the Hidden Profile Paradigm

The Hidden Profile paradigm assesses collective reasoning under distributed information, where no single member has all the facts and success depends on integrating partial knowledge (Fig. 1 and Table 1). While widely applied in human studies, adapting it for LLMs requires formalizing the task structure, information distribution, and success criteria. In this section, we provide that formalization as the basis for controlled experimentation and reproducible benchmark construction.

Let $N$ be the number of agents, indexed by $i = 1, \ldots, N$, and let $\mathcal{O} = \{o_1, o_2, \ldots, o_K\}$ be the set of $K$ possible decision options, among which there is a unique correct option $o^* \in \mathcal{O}$. The full set of task-relevant information $\mathcal{I}$ is divided into shared information $\mathcal{I}_s \subset \mathcal{I}$, available to all agents, and unshared information $\mathcal{I}_u = \mathcal{I} \setminus \mathcal{I}_s$, distributed so that each agent $i$ receives a unique subset $\mathcal{I}_i^u \subset \mathcal{I}u$ with $\bigcup i = 1^N \mathcal{I}_i^u = \mathcal{I}_u$. Each agent's initial knowledge is $I_i = \mathcal{I}_s \cup \mathcal{I}_i^u$. Before communication, agent $i$ makes a *pre-discussion decision* $d_i^{\text{pre}} = f(I_i)$. Agents then exchange messages $M$ over $T$ rounds of communication, after which each makes a *post-discussion decision* $d_i^{\text{post}} = f'(I_i, M)$.

The *Hidden Profile condition* holds when the correct decision cannot be derived from any private information set alone, but becomes attainable once distributed knowledge is pooled through communication: $\exists i$ such that $d_i^{\text{pre}} \neq o^*$ and $f'\left(\bigcup_{i=1}^N I_i, M\right) = o^*$.

To evaluate collective reasoning, we aggregate post-discussion decisions as accuracy using a group rule $A$: $Y^{\text{post}} = A(d_1^{\text{post}}, \ldots, d_N^{\text{post}})$. We consider two rules: the average rule, which measures the proportion of agents selecting the correct option (our default measure of accuracy), and the majority rule, which records whether more than half of the agents select the correct option.

We compare the *Hidden Profile post-discussion accuracy* $Y^{\text{post}}$ against three reference points:

- *Hidden Profile pre-discussion accuracy*: $Y^{\text{pre}} = A(d_1^{\text{pre}}, \ldots, d_N^{\text{pre}})$, providing a baseline for the effect of communication $M$.

- *Full Profile pre-discussion accuracy:* $Y^{\text{full}} = A(d_1^{\text{full}}, \ldots, d_N^{\text{full}})$, where $d_i^{\text{full}} = f(\mathcal{I})$. This serves as an upper bound on individual reasoning, since each agent is given access to the entire information set $\mathcal{I}$.

- *Human group accuracy*: $Y_H = A(d_{h_1}, \ldots, d_{h_N})$, allowing direct comparison between LLM-agent groups and human groups under identical task conditions.

These references allow us to quantify the failure modes of multi-agent LLMs in scenarios where successful information integration is essential, as well as to empirically evaluate whether a task satisfies the Hidden Profile condition. Tasks with low Full Profile pre-discussion accuracy (e.g., $< 80\%$) are unsolvable or too difficult even for individual reasoning, while tasks with high Hidden Profile pre-discussion accuracy (e.g., $> 20\%$) fail to distribute information adequately across individuals. We apply these criteria in automated benchmark construction (Sec. 4.2).

## A.3. Human Group Comparison

For comparison with LLMs, we conducted human-subject experiments with 96 participants (24 groups of four) recruited on Prolific (Palan & Schitter, 2018) in March, April, and August 2025. Groups were assigned to one of three task scenarios (e.g., Table 1), yielding 8 sessions per scenario (24 sessions in total). When randomly assigned to the Hidden or Full Profile condition, participants received asymmetric $I_i$ as in the LLM setup.

Each participant first submitted a pre-discussion decision $d_i^{\text{pre}}$, then engaged in a 15-minute group chat, and finally submitted a post-discussion decision $d_i^{\text{post}}$. Participants earned \$1 for a correct final answer and another \$1 if their group unanimously chose correctly. The study was approved by an Institutional Review Board.

## A.4. Prompts and Communication Templates

**System prompt for multi-agent discussion**

```
%description%

You have received the following information, notice the order of these information are
    randomly shuffle, the order of facts does not indicate importance or relationship,
    please reason carefully:
%information%

Keep your response concise-just one or two sentences. %extra%
```

**User prompt for multi-agent discussion if first to speak**

```
You are the first to speak.
```

**User prompt for multi-agent discussion if not first to speak**

```
Previous messages from other people:
%messages%
It's your turn to speak. %extra%
```

**User prompt for pre-discussion voting**

```
Please decide and provide your rationale in the following JSON format:
{
    "vote": <A string, %possible_answers%>,
    "rationale": <A string, representing your rationale>
}
```

**User prompt for post-discussion voting**

```
Previous messages from other people:
%group_discussion%
Please decide and provide your rationale in the following JSON format:
{
    "vote": <A string, %possible_answers%>,
    "rationale": <A string, representing your rationale>
}
```

**System prompt for automatically generating Hidden Profile tasks**

```
What you're building

Create a group decision task where:
- Everyone sees the same scenario and shared facts.
- Each participant also gets one unique hidden fact that no one else has.
- If people rely only on the shared facts plus their own single hidden fact, they'll
    be pulled toward a specific wrong option.
- Only by sharing all hidden facts can the group see that one option is definitely
```

```
    correct and the others can't be right.

Output format (match this structure)
- name: A string, representing the name of the task.
- description: A short scenario everyone sees.
- shared_information: A list of facts everyone starts with.
- hidden_information: A list with one item per participant. (If you have 4
    participants, include 4 hidden items-one per person.)
- possible_answers: The set of choices to pick from (include at least three).
- correct_answer: The single correct choice (must be one of the options).

Design rules (must all be true)
- At least three options. Exactly one is correct.
- One hidden item per participant. No item is duplicated; each goes to exactly one
    person.
- Shared info is misleading on its own. It should naturally point the group toward a
    particular decoy (a wrong option).
- Shared info + any single hidden item still misleads. If a participant considers only
     the shared info and their own hidden item, the decoy should still look best.
- All hidden items together reveal the truth. When the group pools every hidden item,
    the decoy clearly fails and the correct answer is the only choice that fits all
    facts.
- Every hidden item matters. If you remove any one hidden item, the correct answer
    should no longer be uniquely identifiable.

Step-by-step recipe
1. Pick the basics.
   - Choose the number of participants.
   - Choose at least three options and decide which one is correct.
   - Choose one decoy option you want the shared info to favor at first.
2. Write the shared information.
   - Include solid, plausible facts that make the decoy look like the best choice
    before any sharing happens.
   - Avoid giving away the correct answer here.
3. Create the hidden items (one per participant).
   - Each hidden item should be credible and different from the others.
   - No single hidden item should be enough to prove the correct answer by itself.
   - Across all hidden items, include the decisive details that:
   - Disqualify the decoy from multiple angles, and
   - Show why the correct answer is the only one that satisfies everything.
4. Do the three checks (and revise if needed).
   - Solo check: For each participant, ask: "With only the shared info and this person'
    s hidden item, which option looks best?" It should be the decoy, not the correct
    answer.
   - Group check: With the shared info and all hidden items combined, only the correct
     answer should still make sense; every other option should clash with at least one
    fact.
   - Missing-piece check: Remove any one hidden item and confirm the correct answer is
     no longer uniquely determined.
```

```
An example task:
{
    "name": "evacuation_west_city",
    "description": "You are participating in a study, acting as a community leader of
    a small village surrounded by mountains and rivers. Most villagers own cars, but
    there are also elderly people and children who may need additional assistance when
    walking. Earlier today, heavy rain began to fall, and the local government issued a
     warning about a potential disaster.
Hours ago, you requested relief supplies, but the supply truck has yet to arrive. Now,
```

```
        the rain has temporarily stopped, giving you and the other three community leaders
           a short window to decide on the safest evacuation route before the rain resumes.
          You don't know how much time you have left to make this critical decision.
    Your Task:
    You will discuss with three other participants, who are also acting as community
           leaders, to decide where to evacuate. You have three options:
    - West City: Accessible through a bridge over the river.
    - East Town: Accessible through a tunnel on middle ground.
    - North Hill: Accessible through a driveway and walking trails.
    Usually, it takes the same time to reach all three places by car, but some routes may
          be inaccessible now.
    There is only one correct evacuation location. After the discussion:
    - If you choose the correct location, you will earn $1.
    - If all other participants also choose the correct location, you will earn an
           additional $1 (for a total of $2).
    This means that coordinating with others is critical to maximize your rewards. The
           chat will at most take 15 minutes. However, the exact time when the chat will end
           is unknown.",
        "shared_information": [
            "The local government announced that hotels in West City are prepared to
       accommodate evacuees. While these hotels are fully stocked with food, they may lack
        medical supplies.",
            "The mayor of East Town has offered accommodations for any evacuees. She also
       ensures that volunteers are available to assist them.",
            "The school at North Hill can serve as a temporary evacuation center,
       providing a two-week supply of essentials and sleeping space in the gym.",
            "The river level is still below the bridge to West City."
        ],
        "hidden_information": [
            "The supply truck headed to the village from East Town was stuck in the tunnel
       .",
            "A massive fire has blocked the supply truck and all other traffic.",
            "The walking trails have been closed since last weekend due to fallen trees.",
            "Several villagers reported that a mudslide just occurred, covering the
       driveway to North Hill."
        ],
        "possible_answers": [
            "West City",
            "East Town",
            "North Hill"
        ],
        "correct_answer": "West City"
    }

    In this example, when participants see the description, the shared information and one
           piece of hidden information, they will select a wrong answer. But when they see
          all the information, they will see that the massive fire has blocked the way to
          East Town, and the walking trails and driveway to North Hill both are inaccessibile
          , making West City the only valid option.
```

```
Practical tips
- Think like a mystery: the shared info sets up a convincing-but wrong-first
      impression. The hidden items are the clues that overturn it only when combined.
- Keep each hidden item short and precise (one clear fact per item).
- Avoid redundancy: each hidden item, or the combination of two items, should rule out
      or confirm something different.
- In your notes, make a quick elimination table (rows = facts, columns = options).
      Mark which options survive each fact. By the end, only the correct option should
      survive all rows.
```

```
- If someone sees the description, all shared and all hidden facts, they should
    identify the correct answer before any discussion.
- If someone sees only the description, the shared facts plus one hidden fact, they
    should not be able to identify the correct answer before discussion.

Create one new task. Respond in the following format:
{
    "rationale": <A string representing your rationale for desiging this task. Think
    step by step: think about the case where participants can see the complete
    information, and the cases where they can only see the description, the shared
    information and one piece of hidden information. If someone sees the description,
    all shared and all hidden facts, they should identify the correct answer before any
     discussion. If someone sees only the description, the shared facts plus one hidden
     fact, they should not be able to identify the correct answer before discussion.>
    "name": <A string, representing the name of the task>,
    "description": <A string, representing the description of the task>,
    "shared_information": [
        <A string, representing a piece of shared information>,
        ...
    ],
    "hidden_information": [
        <A string, representing a piece of hidden information>,
        ...
    ],
    "possible_answers": [
        <A string, representing a possible answer>,
        ...
    ],
    "correct_answer": <A string, representing the correct answer>
}
```

**Communication template for discussion**

```
Person N1: %Message N1%
Person N2: %Message N2%
Person N3: %Message N3%
```

