# OpenReview forum: "Systematic Failures in Collective Reasoning under Distributed Information in Multi-Agent LLMs"
_ICML.cc/2026/Conference — ICML 2026 regular_

### Official Review · Reviewer_3n3x · 2026-03-06

**Soundness:** 3
**Presentation:** 3
**Significance:** 2
**Originality:** 3
**Overall Recommendation:** 4
**Confidence:** 3

**Summary:**

This paper studies whether failures in multi-agent LLM systems arise from individual reasoning limitations or from distributed information settings. This paper's main contribution is the introduction of HIDDENBENCH, a benchmark based on the Hidden Profile paradigm that isolates collective reasoning under distributed information from individual reasoning ability. The authors also propose a scalable pipeline to generate such tasks and construct a benchmark with 65 scenarios. Experiments across multiple frontier LLMs show that multi-agent systems perform poorly when critical information is distributed across agents, despite strong single-agent performance with full information. The results reveal systematic failures in information exploration and disclosure in multi-agent interactions.

**Compliance With Llm Reviewing Policy:**

Affirmed.

**Final Justification:**

My concerns are largely resolved, particularly due to the additional discussion and experiments clarifying the generality of the task setting and the effectiveness of structured communication. Then, I raise my score.

**Key Questions For Authors:**

- How do the authors expect the results to change if shared information were neutral or only mildly misleading rather than strongly biased toward a decoy?

- Can the authors provide more concrete real-world examples where the benchmark structure closely matches practical multi-agent deployments?

- What concrete design directions do the authors see as most promising for operationalizing epistemic exploration?

**Limitations:**

yes

**Strengths And Weaknesses:**

### Strength

- Clearly separates whether failures in multi-agent systems arise from reasoning limitations or from distributed information settings.

- Provides a reproducible pipeline for generating Hidden Profile tasks.

- The experimental results clearly demonstrate limitations of current LLM-based multi-agent systems (e.g., Figure 3).

- Analyzes success and failure cases from multiple perspectives, such as summarization and forced information sharing.


### Weakness

- The benchmark follows the Hidden Profile paradigm, where shared information strongly favors an incorrect option. It is unclear how well the findings generalize to broader distributed-information settings.

- The motivation would benefit from clearer real-world examples where this type of information asymmetry naturally arises.

- While the paper diagnoses a failure mode (lack of epistemic exploration), it provides limited guidance on concrete mechanisms (e.g., communication protocols or training objectives) to address it.

---

> ### Author Rebuttal · Authors · 2026-03-30
>
> We thank the reviewer for the clear and constructive feedback.
>
> ### W1: Generalizability beyond Hidden Profile
>
> We clarify that the Hidden Profile paradigm is not a narrow or pathological case, but a minimal, controlled model of a broader coordination challenge: reasoning under distributed information where agents lack meta-knowledge about who knows what.
>
> Its structured design is intentional. The shared-information bias makes failures of information surfacing directly measurable, allowing us to separate coordination failures from confounds such as task difficulty or individual reasoning ability. In more open-ended multi-agent settings, it is often difficult to evaluate whether communication has successfully surfaced and integrated distributed information—failures are easily confounded with ambiguity, retrieval, or task underspecification. In this sense, HiddenBench functions as a **diagnostic stress test** for collective reasoning under distributed information.
>
> **Re: Key Question 1** (neutral/mild asymmetry). If shared information were neutral or only mildly misleading, individual pre-discussion accuracy would rise by design, making failures of information sharing harder to isolate. Our goal is not to model every distributed-information setting, but to construct a controlled benchmark in which this coordination bottleneck is directly testable.
>
> **Additional experiments** show the tasks are not intrinsically unsolvable: on 18 HiddenBench tasks (3 manually crafted \+ 15 randomly sampled), forced disclosure raises accuracy to 92.59% (GPT-4.1) and 98.15% (Gemini-2.5-Flash). The misleading shared information remains present—what changes is that unshared information is surfaced. This confirms the bottleneck lies in the communication process, not task difficulty or inherent deceptiveness.
>
> ### W2: Real-world relevance
>
> We agree that stronger real-world grounding improves the paper and will expand this in the revision. Hidden Profile structure appears in many settings where information is naturally distributed:
>
> - **Medical diagnosis**: different specialists observe different signals; shared symptoms dominate while critical unique findings go unreported.
> - **Incident response** (IT/security): logs, endpoints, and access traces are distributed across teams; the full picture emerges only when teams share unique observations.
> - **Intelligence analysis**: agencies hold complementary partial evidence; failures of information sharing are well documented (e.g., the 9/11 Commission Report).
>
> These settings mirror multi-agent LLM architectures where specialized agents (e.g., researcher, coder, reviewer) operate on partial context and coordinate via message passing. We will clarify this connection in the revision.
>
> ### W3: Limited guidance on concrete mechanisms
>
> We agree that the original submission emphasized diagnosis more than actionable follow-ups. To address this, **we conducted an additional ablation study** testing whether HiddenBench can guide coordination design.
>
> We evaluate a simple structured communication protocol targeting the two dominant failure modes identified by our ablations (failure to surface unshared information, and premature convergence):
>
> - **Exchange** (2 rounds): each agent shares 1–2 decision-relevant facts and gives one reason the current frontrunner could be wrong.
> - **Decide** (1 pass): each agent summarizes the strongest evidence and remaining uncertainty before voting.
>
> This protocol does not explicitly notify agents of information asymmetry and does not require full disclosure.
>
> On 18 HiddenBench tasks (3 manually crafted \+ 15 randomly sampled; 5 runs each, 4 agents):
>
> | Model | Before | After | Improvement |
> | :---- | :---: | :---: | :---: |
> | GPT-4.1 | 0.037 | 0.800 | **\+0.763** |
> | Gemini-2.5-Flash | 0.173 | 0.727 | **\+0.554** |
> | Gemini-2.5-Flash-Lite | 0.043 | 0.743 | **\+0.700** |
>
> This protocol is not intended as a complete solution—it imposes a rigid interaction structure and does not capture real-world communication dynamics. Instead, these results show that HiddenBench is not only diagnostic but also **actionable**: it can identify and evaluate coordination structures that substantially improve collective reasoning under distributed information.
>
> **Re: Key Question 3** (promising design directions). Beyond the structured protocol above, we see several promising directions:
>
> - Structured interaction protocols that enforce evidence sharing before convergence.
> - Anti-convergence mechanisms that delay premature agreement (our dissent ablation is a first step).
> - Training objectives that reward information gain rather than agreement.
>
> Together, these clarifications and additional experiments strengthen the paper along the key dimensions raised: broader benchmark positioning, clearer real-world relevance, and more concrete guidance for improving multi-agent coordination under distributed information. We will revise the paper accordingly.

---

> > ### Author Rebuttal · Reviewer_3n3x · 2026-04-02
> >
> > Thank you for the clear and constructive response. My concerns are largely resolved, especially with the additional discussion and experiments clarifying the role of information sharing and the effectiveness of structured communication. Based on this, I will raise my score at the final decision stage.

---

> > > ### Author Response · Authors · 2026-04-06
> > >
> > > Thank you for your feedback and your decision to raise the score! We’re very pleased that our responses, discussions, and experiments helped address your concerns. We look forward to your updated score at the final decision stage.

---

### Official Review · Reviewer_HMSN · 2026-03-11

**Soundness:** 3
**Presentation:** 3
**Significance:** 3
**Originality:** 3
**Overall Recommendation:** 4
**Confidence:** 5

**Summary:**

This paper investigates reasoning failures of multi-agent LLM systems in distributed information. The paper introduces HiddenBench, a 65-task benchmark (3 manually designed, 5 adapted from prior psychology studies, and 57 generated by GPT-4.1), grounded in the Hidden Profile paradigm. In these tasks, shared information misleads agents toward a wrong answer while unshared information collectively reveals the correct one. The authors evaluate 15 LLMs, suggesting a gap between multi-agent performance under distributed information and single agent performance with complete information. They attribute this failure to the LLM's inability to recognize and act on information asymmetry.

**Compliance With Llm Reviewing Policy:**

Affirmed.

**Key Questions For Authors:**

As listed in the Weaknesses.

**Limitations:**

The authors should discuss limitations as I mentioned in the Weaknesses.

**Strengths And Weaknesses:**

## Strengths

- The research question is well-motivated and timely. Multi-agent LLM systems are increasingly grabbing the attention of the academic and general public (e.g., the explosive Moltbook) and believed as one potential approach toward AGI (so called "distributional AGI" proposed by DeepMind). Understanding their collective failures is important.

- I do appreciate the inclusion of human subject experiments as validation baseline. And the theoretical grounding in the Hidden Profile paradigm provides a reasonable way to disentangle individual reasoning ability from collective coordination failure.

- The structured dissent ablation shows that a single adversarial agent nearly doubles performance, which is a practically interesting result that points toward concrete design directions.


## Weaknesses

- The task filtering criteria might introduce a circularity problem that partially pre-determines the paper's conclusions. By selecting only tasks where pre-discussion Hidden Profile accuracy falls below 20%, the benchmark exclusively retains scenarios where shared information is maximally deceptive. This does not merely ensure that information sharing is required for success, but also ensures agents are actively misled from the outset. The observed collective failures are therefore at least partially a demonstration of LLM susceptibility to misleading shared prompts rather than a pure test of coordination under information asymmetry. I would suggest the authors validating milder asymmetry thresholds or add discussions of this limitation.

- If both humans and LLMs systematically fail these specific tasks, the failure may be intrinsic to the pathological structure of the task rather than diagnostic of reasoning failures. Consider this: why does human collaboration succeed in real-world distributed information settings, such as medicine, law, or engineering, though fails in Hidden Profile tasks? The authors could discuss the boundary conditions under which collective intelligence works versus fails.

- In the ablation experiment, critically, regardless of the prompting strategy, communication depth, or group composition, the multi-agent system consistently underperforms a single agent given complete information. In a way, the trivially optimal solution is simply to route all information to agents before discussion begins, collapsing the multi-agent problem into a single-agent one. A benchmark where the performance ceiling is defined by a non-multi-agent architectural choice offers limited actionable guidance for advancing multi-agent architectures. This risks reducing HiddenBench to a demonstration of a known limitation rather than a scaffold for measurable progress.

---

> ### Author Rebuttal · Authors · 2026-03-30
>
> We thank the reviewer for the thoughtful critique and address each point below.
>
> ### W1: Task filtering circularity
>
> The ≤20% threshold isolates the capability of interest: whether communication can recover correct decisions when no individual agent can do so alone, ensuring that post-discussion improvement reflects information pooling rather than chance.
>
> The reviewer raises an important question: does the benchmark mainly capture susceptibility to misleading shared information, rather than a broader coordination failure? We believe our evidence points to the latter, for three reasons:
>
> **1\) Forced disclosure nearly eliminates failure.** In **additional experiments**, we extended Reveal-All to 18 HiddenBench tasks (3 manually crafted \+ 15 randomly sampled; 54 runs per model-condition pair):
>
> | Model | Reveal-All Accuracy |
> | :---- | :---: |
> | GPT-4.1 | **92.59%** |
> | Gemini-2.5-Flash | **98.15%** |
>
> The misleading shared information remains present—what changes is that unshared information is surfaced. If the primary issue were susceptibility to misleading cues, performance should remain poor after disclosure. Instead, agents recover once the missing information is available.
>
> **2\) Structured communication substantially improves performance.** In a **separate additional experiment**, a simple structured communication protocol (described under W2) targeting the identified bottlenecks raises GPT-4.1 from 0.037 → 0.800 on the same 18 tasks, without changing the underlying information. This suggests the core issue is how communication unfolds—whether agents surface and prioritize unshared evidence—rather than task deceptiveness per se.
>
> **3\) The same pattern is well-established in human groups.** A large body of social psychology literature interprets Hidden Profile failures as coordination failures in information sharing, not simple susceptibility to misleading cues. Our human experiment shows the same qualitative pattern.
>
> We agree that the degree of asymmetry is an important design dimension and will expand the discussion of milder asymmetry settings in the revision.
>
> ### W2: Task structure vs. broader reasoning failure
>
> The reviewer asks: if both humans and LLMs often fail these tasks, does the benchmark reflect something pathological about the task structure rather than a meaningful limitation?
>
> We view the Hidden Profile paradigm not as a pathological corner case, but as a controlled testbed for collective reasoning under distributed information, where communication alone does not guarantee that uniquely held information will be surfaced and integrated.
>
> Where collective intelligence succeeds in practice, additional mechanisms are typically present—structured interaction, role differentiation, or meta-knowledge about who knows what. Current multi-agent LLM systems lack these supports by default.
>
> To test whether performance can improve within the same setting, we **evaluated a new structured communication protocol** that encourages agents to surface decision-relevant facts and uncertainty before convergence, without revealing the information asymmetry or requiring exhaustive disclosure.
>
> On 18 HiddenBench tasks, this protocol substantially improves performance across model families (GPT-4.1: 0.037 → 0.800; Gemini-2.5-Flash: 0.173 → 0.727; Gemini-2.5-Flash-Lite: 0.043 → 0.743). This suggests the failure is not intrinsic to the task structure, but to the absence of coordination mechanisms that help surface and integrate distributed information.
>
> We will make these boundary conditions clearer in the revision.
>
> ### W3: "Single-agent ceiling" / why not centralize?
>
> We view the Full Profile condition as serving a different role than the reviewer's framing suggests.
>
> The Full Profile condition is a **diagnostic reference**, not a deployment recommendation. Its purpose is to separate two failure sources: (1) failures of *information integration* once information is available, versus (2) failures of *information surfacing* through communication. That distinction is central to the paper's contribution.
>
> In practical distributed settings, agents often do not know what information is relevant or who possesses it—even identifying relevant information may require interaction. HiddenBench provides a controlled version of this challenge: all relevant information exists somewhere in the group, but must be surfaced and prioritized through communication.
>
> We will clarify in the revision that the Full Profile condition localizes the bottleneck, not that centralized full-information reasoning is the intended solution.
>
> Together, these clarifications and additional experiments strengthen our interpretation that HiddenBench captures a meaningful coordination bottleneck—one that is measurable, theoretically grounded, and actionable for improving multi-agent collective reasoning.

---

> > ### Author Rebuttal · Reviewer_HMSN · 2026-04-01
> >
> > Thanks for the new evaluations. I'd maintain my positive evaluation.

---

> > > ### Author Response · Authors · 2026-04-06
> > >
> > > Thank you for your positive evaluation! We hope our work can make a meaningful impact on the community.

---

### Official Review · Reviewer_s4Tj · 2026-03-13

**Soundness:** 3
**Presentation:** 3
**Significance:** 2
**Originality:** 2
**Overall Recommendation:** 4
**Confidence:** 4

**Summary:**

The paper studies whether multi-agent LLM systems can solve problems that require combining information distributed across agents. To test this, authors introduce a 65-task benchmark HIDDENBENCH, which is based on the Hidden Profile paradigm from social psychology. This setup is interesting and useful because it separates failures of collective multi-agent reasoning from ordinary single-agent reasoning difficulty.

The paper evaluates 15 models under hidden-information and full-information conditions. The main reported empirical result is a large and consistent gap: distribution does not supprot effectivenes of information processing by LLM agents. The paper’s interpretation is that the limitation is not basic inference, but weak information exchange during interaction.

The submission provides a range of ablation studies to confirm this. Failures persist across prompting styles, communication depth, and group size.

 Overall, the paper’s main claim is to frame collective reasoning in LLM groups as a coordination problem (rather than simply a scaling problem).

**Compliance With Llm Reviewing Policy:**

Affirmed.

**Final Justification:**

Thank you for your rebuttals. My overall weak positive evaluation remains unchanged.

**Key Questions For Authors:**

1. How robust is the main “information surfacing” diagnosis beyond GPT-4.1 and the three tasks used for the deeper ablations?

2. How robust is the main coordination diagnosis beyond the small set of GPT-4.1 ablations?

3. Are there any identified cases where the Hidden Profile framing may be too restrictive or may miss other important forms of multi-agent failure?

**Limitations:**

yes

**Strengths And Weaknesses:**

1. Soundness

The framing is clear and well-founded. The use of full-profile accuracy as a baseline upper bound, along with low pre-discussion accuracy for hidden profiles, is a great design decision.

The empirical foundations are also sound. The paper tests 15 models on the frontier, includes human and GPT-4.1 verification, and has some useful ablations for prompting, depth, group size, dissent, and forced disclosure.

One of my concerns is that some ablations are limited to GPT-4.1 and a few tasks, making the more general diagnosis convincing but not airtight. The empirical footprint is a bit too small for a benchmark paper.

2. Presentation

The paper is well written and easy to follow. The motivation is intuitive. The Hidden Profile setup is presented accessibly.

The paper ties in with the multi-agent LLM literature and the social-psychology literature. It makes the benchmark problem feel well-motivated instead of just being another task suite.

My main issue with the paper is that the claims are a bit sharper than the discussion about them. The benchmark problem, the diagnosis problem, and the coordination problem could be separated a bit more.

3. Significance

The problem is significant. There are a number of reasons to think that the justification for multi-agent LLM systems is the ability to leverage distributed knowledge, and so the evaluation of that ability is important.

The problem has a practical application. The result shows that more agents or discussion is not the answer, and that is a practical application.

The paper is more a diagnosis and evaluation than a solution.

The paper should use/expand ideas more from the "classical" papers/books from area of distributed knowledge representation and knowledge management.

4. Originality

The originality is mainly in the application of the Hidden Profile paradigm for the evaluation of multi-agent LLMs. This is well thought out and makes it possible to disambiguate failures and limits of reasoning.

The construction of the benchmarks has some originality. The mix of adapted human tasks, scenarios, and thresholds for validation is useful for reusing this work.

However, there is no new coordination mechanism, learning algorithm, or theory in this article. The originality is rather related to the formulation of the problem and the experiment.

---

> ### Author Rebuttal · Authors · 2026-03-30
>
> We thank the reviewer for the balanced and constructive feedback. We will revise the paper to more clearly separate three contributions: (i) HiddenBench as a benchmark, (ii) diagnosis of a coordination bottleneck, and (iii) implications for coordination design.
>
> ### W1–W2: Limited ablations and sharp claims
>
> We appreciate this concern. To test whether the main diagnosis generalizes beyond GPT-4.1, **we conducted additional experiments** that substantially broaden the empirical footprint.
>
> **1\) Reveal-All and Secretary** on 18 HiddenBench tasks (3 manually crafted \+ 15 randomly sampled; 54 runs per model-condition pair):
>
> | Model | Reveal-All | Secretary |
> | :---- | :---: | :---: |
> | GPT-4.1 | **92.59%** | 24.07% |
> | Gemini-2.5-Flash | **98.15%** | 71.30% |
>
> Forced disclosure nearly eliminates failure for both models, confirming that **information surfacing—not reasoning—is the primary bottleneck**. The Secretary condition reveals a model-family difference: Gemini-2.5-Flash benefits from passive summarization while GPT-4.1 does not, suggesting integration ability also varies across models.
>
> **2\) Heterogeneous dissent** across 4 additional models (beyond GPT-4.1):
>
> - Gemini-2.5-Flash: Goldilocks pattern (matching GPT-4.1)
> - GPT-4.1-Mini / GPT-5-Mini: improve with more adversarial agents
> - Gemini-2.5-Flash-Lite: robust across compositions
>
> These results suggest the coordination failure is robust across models, while the most effective intervention is model-dependent. We will make this distinction explicit and temper claims accordingly.
>
> ### W3–W5: More diagnosis than solution
>
> We agree. To address this, **we tested whether HiddenBench can support coordination design** by evaluating a structured communication protocol targeting the two dominant failure modes: (1) failure to surface unshared information, and (2) premature convergence.
>
> The protocol introduces two lightweight stages:
>
> - **Exchange** (2 rounds): each agent shares 1–2 decision-relevant facts and gives one reason the current frontrunner could be wrong.
> - **Decide** (1 pass): each agent summarizes the strongest evidence and remaining uncertainty before voting.
>
> **The protocol does not notify agents of asymmetry or require full disclosure.**
>
> On 18 HiddenBench tasks, this yields large gains across model families:
>
> | Model | Before | After | Improvement |
> | :---- | :---: | :---: | :---: |
> | GPT-4.1 | 0.037 | 0.800 | **\+0.763** |
> | Gemini-2.5-Flash | 0.173 | 0.727 | **\+0.554** |
> | Gemini-2.5-Flash-Lite | 0.043 | 0.743 | **\+0.700** |
>
> This protocol is not a complete solution—it imposes a rigid structure and omits real-world communication dynamics. However, these results strengthen the paper beyond "benchmark \+ failure mode": HiddenBench can serve as a **development testbed** for coordination mechanisms that improve LLM collective reasoning. We will include this experiment in the revision.
>
> ### W4: Classical literature
>
> We will expand the related work to engage more directly with classical literature on distributed knowledge and coordination, including **Distributed epistemic logic** (Fagin et al., 1995), **Transactive memory** (Wegner, 1987), and **Exploration–exploitation** (March, 1991). This framing will clarify that the benchmark is grounded in a broader literature on collective reasoning, not only in recent multi-agent LLM work.
>
> ### Q1–Q2: Robustness of the diagnosis beyond GPT-4.1
>
> Addressed above. The Reveal-All / Secretary analysis now spans GPT-4.1 and Gemini-2.5-Flash on 18 scenarios, and heterogeneous-dissent covers five models total. Together, these confirm that the main diagnosis—failure to surface critical unshared information—is not specific to a single model or prompting setup.
>
> ### Q3: Is the Hidden Profile framing too restrictive?
>
> Our target is collective reasoning under distributed information. A central evaluation challenge is that failures in open-ended multi-agent settings are confounded with ambiguity, retrieval, or task underspecification.
>
> We adopt the Hidden Profile paradigm precisely because it isolates this coordination failure: groups fail to surface and integrate uniquely held information even when doing so is necessary. This provides a clear accuracy-based criterion—whether communication enables recovery of the correct answer—difficult to obtain in open-ended settings.
>
> The paradigm has been used in social psychology not only to document failures but to study how collective reasoning improves—through structured discussion, dissent roles, and decision protocols. We view HiddenBench similarly: a testbed for studying when and how coordination succeeds or fails. Our structured-communication results directly illustrate this.
>
> We position HiddenBench as a structured, theory-grounded benchmark for one foundational coordination regime—whether communication successfully surfaces and integrates distributed information—not a complete account of multi-agent reasoning failure. We will clarify this scope in the revision.

---

> > ### Author Rebuttal · Reviewer_s4Tj · 2026-04-03
> >
> > Thank you for the responses. My overall positive evaluation remains unchanged.

---

> > > ### Author Response · Authors · 2026-04-06
> > >
> > > Thank you for your positive evaluation! We hope our work can make a meaningful impact on the community.

---

### Official Review · Reviewer_bYot · 2026-03-14

**Soundness:** 3
**Presentation:** 3
**Significance:** 2
**Originality:** 3
**Overall Recommendation:** 4
**Confidence:** 4

**Summary:**

The paper evaluates the ability of multi-agent large language models (LLMs) to perform collective reasoning when information is distributed across agents. The main contribution is to the introduction of HIDDENBENCH, a theory-grounded benchmark inspired by the Hidden Profile paradigm.
Using this benchmark, the authors evaluate 15 frontier LLMs in multi-agent settings where each agent holds partial information that must be shared to reach the correct decision. Through extensive experiments and ablations, the paper identifies the core failure mode: agents fail to recognize latent information asymmetry and prematurely converge on shared evidence rather than actively eliciting unshared information from others. These failures persist across prompting strategies, communication depths, and group sizes, and in many cases worsen as the number of agents increases.

**Compliance With Llm Reviewing Policy:**

Affirmed.

**Final Justification:**

I maintain my positive score.

**Key Questions For Authors:**

1）During the task filtering, when the paper states “low accuracy (≤ 20%) in the Hidden Profile condition,” does this mean that the accuracy of each individual agent is below 20%?
2） Please clarify the detailed experimental setup of the “Share All Information” condition in Table 3. Why does this setting still perform substantially worse than the single-agent setting with full information? Could the authors provide further analysis of the underlying reasons for this gap?
3）How is the adversarial agent implemented in Table 5? Please provide a more detailed description of the prompting strategy or mechanism used to realize adversarial behavior.

**Limitations:**

see weakness

**Strengths And Weaknesses:**

Strengths

1 The paper addresses an important and underexplored problem—collective reasoning under distributed information in multi-agent LLM systems—and provides a well-motivated evaluation setting.

2 The proposed HIDDENBENCH benchmark (includes 65 tasks) systematically isolates collective reasoning from individual reasoning ability, enabling controlled and reproducible evaluation of multi-agent coordination failures.

3  The paper evaluates 15 frontier LLMs and conducts multiple ablation studies (prompting strategies, communication depth, group size), offering useful insights into the sources of failure in multi-agent reasoning.


Weakness

1. The paper mainly diagnoses failure modes but does not propose concrete algorithmic methods or strategies

---

> ### Author Rebuttal · Authors · 2026-03-30
>
> We thank the reviewer for the helpful feedback and specific questions.
>
> ### W1: "No concrete algorithmic methods"
>
> We agree that the original submission focused primarily on diagnosis. To address the reviewer’s concern about concrete algorithmic strategies, we have conducted an additional ablation to test whether HiddenBench can be used to identify and evaluate coordination mechanisms for collective reasoning under distributed information.
>
> Specifically, based on the failure modes revealed by the benchmark, we evaluate a simple structured communication protocol with two stages:
>
> - **Exchange** (2 rounds): each agent shares 1–2 decision-relevant facts and gives one reason the current frontrunner could be wrong.
> - **Decide** (1 pass): each agent summarizes the strongest evidence and remaining uncertainty before voting.
>
> Importantly, this protocol does not explicitly notify agents of information asymmetry and does not require full disclosure.
>
> On 18 HiddenBench tasks (3 manually crafted \+ 15 randomly sampled from the benchmark; 5 runs each, 4 agents), this yields substantial improvements:
>
> | Model | Before | After | Improvement |
> | :---- | :---: | :---: | :---: |
> | GPT-4.1 | 0.037 | 0.800 | **\+0.763** |
> | Gemini-2.5-Flash | 0.173 | 0.727 | **\+0.554** |
> | Gemini-2.5-Flash-Lite | 0.043 | 0.743 | **\+0.700** |
>
> These results show that HiddenBench is not only diagnostic but also useful for developing and evaluating coordination strategies: it enables systematic identification of communication structures that substantially improve collective reasoning under distributed information. We will include this experiment in the revision.
>
> ### Q1: ≤20% accuracy criterion
>
> Yes. This refers to **individual** pre-discussion accuracy, averaged across agents and sessions. We run 10 sessions with GPT-4.1 agents and record each agent's decision before any interaction. The ≤20% threshold ensures that success requires information pooling rather than local inference. We will clarify this in the revision.
>
> ### Q2: "Share All Information" vs. Full Profile
>
> **"Share All Information"** \= Partial information \+ instructed disclosure; **Full Profile** \= Complete information \+ no disclosure required. To be more specific:
>
> - **"Share All Information"** (Table 3\) is a *prompting* intervention: agents are instructed to disclose all information they have, but must still do so voluntarily through standard multi-agent dialogue. This contrasts with **"Reveal-All"** (Table 6), which is a mechanistic intervention that appends all information available to an agent to its round-1 message, bypassing the communication bottleneck entirely.
> - **Full Profile** provides all agents with complete information (shared \+ unshared) from the outset, serving as an upper bound that isolates individual reasoning from coordination.
>
> The gap between **"Share All Information"** and **Full Profile** (0.467 vs. 0.733) shows that prompting alone does not produce effective information surfacing. In **"Share All Information"** condition, even when we explicitly prompt agents to share all available information, they often converge early on shared evidence, which biases subsequent communication toward reinforcing the emerging consensus rather than surfacing novel information.
>
> **In additional experiments**, we extended Reveal-All to 18 HiddenBench tasks (3 manually crafted \+ 15 randomly sampled), achieving 92.59% (GPT-4.1) and 98.15% (Gemini-2.5-Flash). This confirms that agents can integrate information once it is surfaced—the bottleneck lies in the communication process, not reasoning. We will clarify this distinction in the revision.
>
> ### Q3: Adversarial agent implementation
>
> The adversarial agent uses the same system prompt and user prompt as the "Very Conflictual" condition in Table 3, which instructs agents to actively challenge claims, oppose emerging consensus, and avoid agreement-seeking behavior. The specific prompt appended is: *"Reject all attempts at agreement. Oppose every claim, dismantle arguments relentlessly, and treat the conversation as a battleground where domination—not dialogue—is the objective."* The difference between Tables 3 and 5 is:
>
> - **Table 3** (homogeneous): *all* agents receive the adversarial prompt.
> - **Table 5** (heterogeneous): only a *subset* of agents receive it, while the rest use the default prompt.
>
> This heterogeneous composition produces the Goldilocks effect reported in the paper: one adversarial agent disrupts premature consensus and improves accuracy, but too many adversarial agents impair the group's ability to converge on any decision.
>
> We hope these clarifications and additional results address the reviewer’s concerns and further strengthen the contribution of the paper.

---

> > ### Author Rebuttal · Reviewer_bYot · 2026-04-03
> >
> > Thanks for the responses. I'd maintain my positive evaluation.

---

> > > ### Author Response · Authors · 2026-04-06
> > >
> > > Thank you for your positive evaluation! We hope our work can make a meaningful impact on the community.

---

### Decision · Program_Chairs · 2026-04-30

**Decision:**

Accept (regular)

**Comment:**

This paper studies whether multi-agent LLM systems can solve problems that require combining information distributed across agents. To test this, the authors introduce a 65-task benchmark called HIDDEN BENCH, based on the Hidden Profile paradigm from social psychology.

The paper evaluates 15 models under hidden-information and full-information conditions. The main reported empirical result is a large and consistent gap: distribution does not support the effectiveness of information processing by LLM agents. The paper’s interpretation is that the limitation is not basic inference, but weak information exchange during interaction. The submission provides a range of ablation studies to confirm this. Failures persist across prompting styles, communication depth, and group size.

Some limitations that the reviewers identified in this submission include the lack of a coordination mechanism, a learning algorithm, or a theory. [Note: There is a related line of work on understanding the transfer mechanisms behind multitasking ([1](https://arxiv.org/abs/2005.00944),[2](https://arxiv.org/abs/2602.03783),[3](https://arxiv.org/abs/2010.11750)).

The paper is reviewed positively by four independent referees. After a productive discussion, all the reviewers recommend weak acceptance. As a result, I would like to follow their recommendations. We encourage the authors to incorporate all rebuttal additions and clarifications, and the above related work discussions into the final version.